# Enriched Alternative Splicing in Islets of Diabetes-Susceptible Mice

**DOI:** 10.3390/ijms22168597

**Published:** 2021-08-10

**Authors:** Ilka Wilhelmi, Alexander Neumann, Markus Jähnert, Meriem Ouni, Annette Schürmann

**Affiliations:** 1Department of Experimental Diabetology, German Institute of Human Nutrition (DIfE), 14558 Potsdam, Germany; Ilka.wilhelmi@dife.de (I.W.); Markus.jaehnert@dife.de (M.J.); meriem.ouni@dife.de (M.O.); 2German Center for Diabetes Research (DZD), 85764 Munich, Germany; 3Omiqa Bioinformatics, 14195 Berlin, Germany; Alexander.neumann@omiqa.bio; 4Institute of Nutritional Sciences, University of Potsdam, 14558 Nuthetal, Germany; 5Faculty of Health Sciences, Joint Faculty of the Brandenburg University of Technology Cottbus-Senftenberg, The Brandenburg Medical School Theodor Fontane and The University of Potsdam, 14469 Potsdam, Germany

**Keywords:** alternative splicing, epigenetic, MicroRNA, RNAseq, diabetes, β-cell failure

## Abstract

Dysfunctional islets of Langerhans are a hallmark of type 2 diabetes (T2D). We hypothesize that differences in islet gene expression alternative splicing which can contribute to altered protein function also participate in islet dysfunction. RNA sequencing (RNAseq) data from islets of obese diabetes-resistant and diabetes-susceptible mice were analyzed for alternative splicing and its putative genetic and epigenetic modulators. We focused on the expression levels of chromatin modifiers and SNPs in regulatory sequences. We identified alternative splicing events in islets of diabetes-susceptible mice amongst others in genes linked to insulin secretion, endocytosis or ubiquitin-mediated proteolysis pathways. The expression pattern of 54 histones and chromatin modifiers, which may modulate splicing, were markedly downregulated in islets of diabetic animals. Furthermore, diabetes-susceptible mice carry SNPs in RNA-binding protein motifs and in splice sites potentially responsible for alternative splicing events. They also exhibit a larger exon skipping rate, e.g., in the diabetes gene *Abcc8*, which might affect protein function. Expression of the neuronal splicing factor *Srrm4* which mediates inclusion of microexons in mRNA transcripts was markedly lower in islets of diabetes-prone compared to diabetes-resistant mice, correlating with a preferential skipping of SRRM4 target exons. The repression of *Srrm4* expression is presumably mediated via a higher expression of miR-326-3p and miR-3547-3p in islets of diabetic mice. Thus, our study suggests that an altered splicing pattern in islets of diabetes-susceptible mice may contribute to an elevated T2D risk.

## 1. Introduction

More than 420 million people worldwide suffer from type 2 diabetes (T2D), meaning that 1 out of 11 adults is affected, and the prevalence is constantly rising. In the pre-diabetic state, pancreatic β-cells can compensate for insulin resistance by increasing β-cell mass and insulin secretion. Obese mice which are hyperglycemic and insulin-resistant, such as New Zealand Obese (NZO) mice, are used as a model system in this context. However, B6-ob/ob mice, which carry a mutation in the *Leptin* gene, become obese and insulin-resistant without developing diabetes, and therefore serve as non-diabetic controls. Working with mouse models of divergent genotypes and different diabetes susceptibility gives insights into genetic marks that are associated with the onset and progression of the disease. Transcriptome analysis of the two mouse strains identified significant differences in expression profiles of genes that are associated with β-cell proliferation, survival and primary cilia function [1,2,3].

While the focus so far was to identify diabetes genes via their differential expression, here we focused on alternative splice events and analyzed RNA-seq data. The majority of coding genes undergo alternative splicing, which enhances the proteome complexity and tremendously increases the genome’s coding capacity [4,5]. Alternative splicing is a tightly regulated process which requires the recognition of splice sites and catalytic reactions of splicing activators and repressors [6]. These splicing regulators are RNA binding proteins (RBPs) such as the trans-acting SR proteins, a family of proteins with a domain enriched for serine (S) and arginine (R) residues, and the heterogeneous nuclear ribonucleoproteins (hnRNPs). On the one hand, SR proteins predominantly bind to short motifs of the RNA in exonic regions and are positive splicing regulators that foster exon inclusion [7]. On the other hand, hnRNPs preferentially bind to intronic sequences and promote exon skipping by steric blocking of splicing factor binding [8,9]. However, there is evidence for a position-dependent effect on RBP function. SR proteins activate splicing only when recruited upstream of the 5’ splice site, whereas hnRNPs can activate splicing only when bound downstream of the 5’ splice site [10]. Mutations in RBP-binding motifs can lead to gain or loss of function of splicing. Binding of exonic repressors enhance splicing when bound to introns and exonic activators can repress splicing when bound to introns [11]. Alternative splicing affects gene expression [12] and alters protein function, including the secretory pathway [13]. About 50–60% of disease-associated mutations are splice defects, and 15% of all inherited diseases are caused by mis-splicing [14]. The group of Decio L. Eizirik, for instance, studied the impact of splicing switches on autoimmune type 1 diabetes and detected alterations in T-cells and lymph node stromal cells as well as in β-cells that were exposed to pro-inflammatory cytokines [15,16,17]. As splicing generally occurs co-transcriptionally, it is conceivable that exon inclusion or exclusion from the mature mRNA is also influenced by epigenetic alterations including histone modifications [18] and DNA methylation [19]. For instance, histone deacetylases (HDACs) influence the selection of specific pairs of splice sites in many genes. In particular, local histone modifications including H3K36me3 and H3K9me3 as well as acetylation label chromosomal regions surrounding alternative exons to influence splicing [20]. 

Here, we present a comprehensive study of alternative splicing in islets of Langerhans of a mouse model for T2D. 

## 2. Results

### 2.1. Differential Alternative Splicing in Islets of Diabetes-Prone and -Resistant Mice

NZO and B6-ob/ob-mice differ in their susceptibility to developing diabetes. The use of a specific feeding regimen consisting of a carbohydrate-free diet for 15 weeks followed by a 2-day carbohydrate challenge represents a diabetogenic stimulus sufficient to induce alterations in gene expression, finally leading to β-cell loss in NZO mice [1,2]. This particular dietary stimulus was applied for the current study (Figure 1A) and led to significantly increased blood glucose levels in NZO compared to B6-ob/ob mice without affecting body weight (Appendix A). RNAseq data from isolated islets of both strains were screened for alternative splicing events with a focus on (i) differential expression of histones and chromatin modifiers, (ii) microexons, (iii) enriched RNA binding protein motifs, and (iv) SNPs located in splicing relevant sequences (Figure 1A). Principal component analysis of the RNAseq dataset revealed consistent gene expression within the two groups, and a high variation between them (Figure 1B). 

The screening for alternative splicing detected 894 significant events (∆PSI > 0.1, probability > 0.9) in 730 genes. Among these, about half of the alternatively spliced genes (*n* = 381) showed a differential expression. According to KEGG pathway enrichment analysis, alternatively spliced genes are enriched in molecular functions such as endocytosis and insulin secretion (Figure 1C, Appendix A). Most alternative splicing events affect cassette exons (*n* = 502), followed by cases of intron retention (*n* = 204) and alternative 3’ (*n* = 98) and 5’ splice sites (*n* = 90), respectively (Figure 1D, donut). The majority of alternatively spliced cassette exons are located within the coding sequence of the gene (347 events in 273 genes) (Figure 1D, open green circle). Skipping of these exons preserves the reading frame in 55% (*n* = 150) of cases, whereas 40% (*n* = 109) result in a frame shift. This is significantly different from the expected rate of frameshift induction upon random exon skipping, which is 66%. Notably, a small number of the splicing events (5%) which might induce a frameshift have multiple alternative exons that can rescue the reading frame if skipped together (*n* = 14). However, only half of the frameshift-inducing events are associated with differential gene expression (Figure 1D, open gray circle). This suggests that most of the alternative splicing events identified in the islets of diabetes-susceptible mice contribute to proteins with altered functions rather than affecting expression levels. 

### 2.2. Altered Expression of Histones and Chromatin Modifiers

Recent studies showed that the chromatin accessibility as well as histone deacetylase (HDAs) activity influence splice site selection [21,22,23]. Therefore, we analyzed the expression pattern of histone and chromatin modifiers in order to evaluate the degree of the chromatin accessibility in islets of the two mouse models. In NZO islets, 22 genes were expressed at higher levels and 54 at lower levels than in B6-ob/ob (Figure 2). As depicted in the heatmap in Figure 2, around 60% of the HDACs and the methyltransferases were downregulated in islets of NZO mice, which could in theory reflect a higher degree of chromatin accessibility. This downregulation might be a potential mechanism to modulate alternative splicing in islets. The lower part of the heatmap depicts the expression pattern of the small nuclear RNA (snRNA) U2 family, components of the spliceosome complex, facilitated by open chromatin structure [20].

### 2.3. Differences in Splicing of Microexons in NZO and B6-ob/ob Islets

A specific type of exon known to retain the reading frame of the mRNA and to influence the protein’s properties is the so-called microexons. These are small exons 3–30 nucleotides in length and are found in brain, heart and muscle cells, whereas a number of microexons are exclusively regulated in the brain [24]. Inclusion of microexons in mRNA transcripts is mediated by a neuronal-enriched splicing factor the serine/arginine repetitive matrix 4 (SRRM4) [25]. Insulin-producing β-cells are described to express neuronal-specific genes [26]. We found that *Srrm4* is expressed in B6-ob/ob islets at similar levels as in neuronal tissue. In contrast, islets of NZO mice exhibit low expression levels, as validated by qRT-PCR, which correlate with exon skipping/inclusion rate in comparison with neuronal and non-neuronal tissues (Figure 3A,B). In line with this, known SRRM4 target exons are preferentially skipped in islets of NZO mice (Figure 3B). The same exons are associated with autism spectrum disorder [27]. Altered microexons are also found in genes encoding for proteins involved in cellular processes like vesicle mediated exocytosis and secretion such as *Dctn1* (Dynactin subunit 1) or *Sptan1* (Spectrin alpha, non-erythrocytic 1) (Appendix A). The latter is involved in regulated Ca^2+^ dependent exocytosis [28], a process also important for insulin release from pancreatic β-cells. Alternative microexons are further found in genes encoding for proteins involved in cell growth and proliferation (e.g., *Ptk2*, *Gnas*), or in the regulation of cell polarity and cell adhesion (e.g., *Mpp7*) (Appendix A). In order to clarify the molecular basis for the lower *Srrm4* expression in NZO islets, we took advantage of our miRNome data from islets of the two mouse models [29] in combination with a computation framework for target prediction [30]. Thereby, we detected two microRNAs, miR-326-3p and miR-3547-3p, which are expressed at a significantly higher level in NZO than in B6-ob/ob islets (Figure 3C), and may target and silence *Srrm4* expression. Taken together, our data support the idea that disrupted splicing patterns of cassette exons and microexons in NZO islets might contribute to a disease-promoting environment and the development of T2D, as suggested earlier for other diseases [31]. 

### 2.4. Role of Neuron-Specific Splicing Factors and Enrichment of Affected RNA Binding Protein Motifs Splicing Relevant

As the neuron-specific splicing factors NOVA1, RBFOX and CELF are significantly enriched in human islets [17], we tested whether this was true for mRNA levels of the murine orthologs. Expression of *Nova1* and *Rbfox2* is significantly lower in NZO islets in comparison to B6-ob/ob (Figure 4). Several NOVA1 target genes are involved in exocytosis, apoptosis, insulin receptor signaling and others [32], indicating that a lower abundance of this splicing factor in NZO islets participates in their β-cell loss. Another important candidate is the diabetes gene *Glis3*, a transcription factor which regulates splicing of the pro-apoptotic BH3 protein Bim. Accordingly, the suppression of *Glis3* increases β-cell apoptosis [33]. In NZO islets, *Glis3* expression is significantly lower than in B6-ob/ob islets (Figure 4).

Splicing changes are orchestrated by regulatory networks [34]. We were interested in whether the differential splicing events in diabetes-susceptible islets are coordinated by specific master splicing factors. Thus, in order to identify possible common regulators of the observed splicing changes, we performed a motif analysis for enriched RBP motifs and investigated the abundance of 47 RBP motifs in and around alternative exons that were identified in NZO islets. The enrichment analysis revealed that the RBP motifs which were significantly associated with exon inclusion in NZO mice were TUT1, SRSF7, PCBP2 (located upstream introns), G3BP2 and SRSF1 (found downstream exons). Exons with lower inclusion levels in NZO islets harbored motifs of SRSF3, PCBP1, HHNRNPK, SRSF10, RBM24, BRUNOL5, HNRNPBH2, and SRSF10 within the exon or flanking introns (Figure 5A). 

In order to evaluate the molecular basis for the high number of different splicing events between diabetes-resistant and diabetes-susceptible mice, we screened for SNPs in the vicinity of alternative exons. For intronic SNPs, sequences 15–500 nucleotides up- and downstream of alternative exons were analyzed. SNPs adjacent to splice sites were supposed to be within 15 nucleotides up- or downstream of splice sites, and exonic SNPs were within the whole exon except for the already mentioned 15 nucleotides close to the splice site. When normalized to a length of 1000 nucleotides, most SNPs were found in the 5’ splice sites. The fewest SNPs were localized within the alternative exon (Appendix A). Overall, NZO mice carry a significantly higher number of SNPs adjacent to alternatively spliced cassette exons when compared to the global SNP frequency (Figure 5B). We therefore hypothesized that these SNPs might lead either directly or indirectly to a change in splicing either by disrupting a splice site or by affecting RBP motifs. Indeed, 86 alternative splicing events which could impair binding of at least one of the RBPs mentioned above were identified. How a loss of binding sites of RBP can influence splicing is shown for *Akr1c12* (Aldo-keto reductase family 1, member C12) (Figure 5C). A SNP in a SRSF1 binding site in the NZO genome might be responsible for around 80% of the reduced exon inclusion in NZO islets. SRSF1 binding sites were identified to be enriched in downstream introns of alternative exons that are more often included in islets from NZO mice (Figure 5A). The total number of exon-skipping events was ~16,000 in NZO mice. Disruption of this motif led to the opposite effect, enhancing exon skipping in NZO mice. However, only very few SNPs leading to enhanced inclusion were identified in islets from NZO mice, in accordance with the observation that NZO mice show more exon skipping in general. Whether the SNPs identified and discussed here directly affect the detected differences in splicing needs further proof. It is also possible that SNPs not located in splicing sites, but in genes encoding for splicing factors, impact splicing.

### 2.5. Polymorphisms in Splice Sites in Diabetes-Prone Mice

Besides the disruption or creation of RBP binding sites, mutations in canonical splice sites can result in splicing defects. Mutations in the 5’ splice site, for instance, often lead to exon skipping [35]. We detected SNPs within or in close proximity to splice sites in 124 genes (Appendix A). Among these are important genes such as *Adgrf5* (Adhesion G protein-coupled receptor F5), also designated GPR116. The hepatokine FNDC4 binds to this receptor and promotes insulin signaling in adipocytes [36]. Similarly, the diabetes gene *Arl15* was described to improve insulin-mediated AKT phosphorylation in myotubes [37]. Short-term inhibition of the autophagy gene *Atg7* in β-cells improved their function [38]. Polymorphisms in the glucokinase-associated dual-specificity phosphatase 12 (*Dusp12*) are associated with T2D [39]. Pathway analysis of the 124 genes with SNPs potentially relevant for splicing indicated an enrichment in cell division processes, signal transduction and protein ubiquitination (Figure 5D, Appendix A), suggesting that genetically mediated splicing alterations can play a role in diabetes development that has been underestimated until now.

As we detected several diabetes-associated genes among those genes which carry SNPs in or in close proximity to the splice sites, we next mapped the 421 diabetes genes discovered by GWAS [40] onto the 730 alternatively spliced mouse genes. We detected 16 genes with differential splicing events in NZO and B6-ob/ob islets (Appendix A). Eleven of these 16 genes carry SNPs that might affect splicing (Table 1). Interestingly, screening for the loss of CpG sites which putatively affects expression by DNA methylation revealed eight genes, in which 1 to 15 CpG sites were deleted. We also screened databases for phenotypes of the corresponding knockout mice of the 11 genes. Eight knockout mice were listed in the MGI and IMPC databases (http://www.informatics.jax.org/phenotypes.shtml, accessed on 12 April 2021; https://www.mousephenotype.org/, accessed on 12 April 2021), and seven (*Abcc8, Aktip, Myt1l, Nrxn1*, *Ppcn2, Tcf7l2,* and *Tpcn2*) exhibited phenotypes with altered metabolic traits (Table 1).

Prominent examples are well-known diabetes genes such as *Tcf7l2* and *Abcc8* (also named *Sur1*). *Abcc8* carries a SNP close to the splice site of exon 34 that might lead to exon skipping (Figure 5E). *Abcc8* was also identified in a network of genes with alternatively spliced cassette exons, generated by Ingenuity Pathway Analysis (IPA) (Appendix A). Although the G to A mutation at the +3 position should promote base pairing with the U1 snRNA and support exon definition [41,42], both exons 33 and 34 of *Abcc8* were almost completely skipped in islets of NZO mice and included in islets of B6-ob/ob mice. Disease-associated G to A mutations were found in other cases and were suggested to act in a context-dependent manner [43]. Besides other changes in the trans-acting environment that regulate the described splicing change, it is possible that the complex recognition of the two consecutive exons is inhibited by changing the second 5’ splice site. Skipping of a single one of these usually constitutive exons would induce a frameshift, whereas simultaneous skipping preserves the reading frame. Indeed, by splice-specific RT-PCR, only the skipping isoform (Δ33Δ34) was detected in RNA from NZO islets (Figure 5G), whereas overall gene expression of *Abcc8* was unchanged (Appendix A). *Abcc8* encodes a subunit of the ATP-dependent potassium channel that is required for glucose-stimulated insulin release from pancreatic β-cells [44,45]. Mutations in the human *ABCC8* gene are associated with T2D and are highly linked to the development of neonatal diabetes. Most of the described mutations in *ABCC8* are reported as missense mutations that affect the function of the potassium channel [46]. Only a few of the known mutations are associated with potential splicing defects. For instance, an SNP within intron 32 should result in the skipping of exon 33 [47]. This induces a frameshift, resulting in a protein with a shortened C-terminus. Such shorter variant was detected in hypothalamus, midbrain, heart and Min6 cells which exhibited altered electrophysiological activities. Skipping of both exons 33 and 34, as detected in NZO islets, would also affect the C-terminus of *ABCC8* and potentially alter the second intracellular nucleotide-binding domain (NBD2). There is a diabetes-associated human SNP (rs137852676) at the splice site of exon 34 in the human orthologue, similar to the SNP we identified in NZO mice. This SNP is reported as missense, but due its position exactly at the splice site, it is likely that it actually alters splicing patterns.

## 3. Discussion

Our systematic analysis of alternative splicing events in islets of Langerhans from two well-established obese mouse models differing in their diabetes susceptibility identified (i) altered expression of histone and chromatin modifiers, which could also affect splicing, (ii) a large number of slicing switches, including those of microexons, (iii) a higher degree of exon skipping in the diabetes-prone mouse, (iv) an enrichment of alterations in binding motifs of specific splicing factors, and (v) polymorphisms in putative splice sites. All these effects most likely contribute to the impaired expression and function of proteins involved in appropriate β-cell function and in the lack of compensation for a higher insulin demand under obese conditions during diabetes development. 

Dysregulated splicing is known to cause disease [35]; however, only limited knowledge exists on defective splicing in islets mainly linked to type 1 diabetes [17]. NZO and B6-ob/ob mice, which were both obese, insulin-resistant and normoglycemic on a carbohydrate-free diet, were fed a carbohydrate-containing diet for two days, resulting in elevated blood glucose levels and strong changes in the islet transcriptome in NZO mice [1,48]. Here, we took advantage of islet RNAseq data at this stage and identified a large number of divergent splicing events, mainly affecting cassette exons and intron retention. NZO islets exhibited a higher degree of exon skipping, which was experimentally confirmed for the diabetes gene *Abcc8*. Frame-preserving splicing changes may modulate protein functionality, e.g., by impairing protein–protein interactions, influencing the three-dimensional structure, changing the subcellular localization of the protein, downstream signaling and others.

Alternative splicing is particularly well-studied in neuronal development and brain function, and dysregulated splicing is linked to neurologic and neuropsychiatric disorders. As islet β-cells show a large overlap in the expression pattern and the secretory machinery of neurons, it is reasonable that we detected several similarities. First of all, NZO islets exhibited a lower level of microexon inclusions, which is at least partially caused by reduced expression of the splicing factor *Srrm4*. SRRM4 regulates most of the microexon splicing events and appears to be suppressed by elevated expression of miR-326-3p and miR-3547-3p, which theoretically target *Srrm4*. Microexons preserve the reading frame and insert only a few amino acids in loop regions of the corresponding protein and they are enriched on the surfaces of protein interaction domains [27]. Besides *Srrm4*, the expression of two other neuron-specific splicing factors, *Nova1* and *Rbfox2*, as well as the diabetes gene *Glis3*, which regulates splicing of *Bim3*, encoding an apoptosis protein, is lower in NZO than in B6-ob/ob islets. We believe that all these alterations contribute to β-cell failure and diabetes development of the NZO mice. 

Genetics are heavily linked to the susceptibility of T2D and the comparison of the NZO and B6-ob/ob genomes lead to the identification of several diabetes genes like *Lefty1*, *Apoa2*, *Pcp4l1* and others [2,48]. Similarly, at least about 120 changes in splicing of islet genes appear to be caused by polymorphisms, because we detected SNPs in or in close proximity to splice sites. This observation is limited to bioinformatic analysis of RNAseq data from only two different mouse strains and needs to be further proven. Nevertheless, the comparison of two mouse strains with different genotypes was already successful to identify a single SNP that has an impact on metabolic health in another context [49]. In addition to this, changes in microRNA expression as demonstrated for miR-326-3p and miR-3547-3p can affect key splicing regulators like *Srrm4*. Comparatively, the role of miRNAs in the regulation of alternative splicing of tau was investigated in the brain, and their changes were shown to associate with the development of Alzheimer’s disease [50]. One example is miR132, which directly targets the neuronal splicing factor PTBP2 (polypyrimidine tract-binding protein 2) [51].

In addition, epigenetic mechanisms, including histone modifications have an impact on alternative splicing. This is comprehensible because nucleosome density is higher over exons than introns [20]. Histone proteins were shown to interact with splicing factors (e.g., Rbfox2, Rbfox3, Sfpq) and changes in histone methylation impact alternative splicing of selected genes [52]. Numerous histone-modifying enzymes showed strong differences in their expression in NZO and B6-ob/ob islets. This can lead to overall differences in the expression pattern, but also in alterations of exon inclusion or exclusion from the mature mRNA. For the myeloid cell leukemia sequence 1 (*MCL1*) transcript for example, alternative splicing of exon 2 generates a protein which either prevents or supports cell death of human cancer cells. Inhibition of HDAC activity markedly reduced the occupancy of exon 2 by RNA polymerase II and some splicing factors [53].

In summary, our comprehensive analysis of islet RNAseq data of diabetes-resistant and -susceptible mice detected an enrichment of alternative splicing, in particular an elevated exon skipping and impaired microexon inclusion in islets from diabetes-susceptible mice. We believe that mis-splicing impairs protein function, and thereby contributes to the onset of diabetes, as is the case for other diseases [35]. Among the genes with affected splicing are well known diabetes genes such as *Abcc8* and *Tcf7l2*. However, further work is needed to support the suggestion that aberrant splicing not only affects islet transcript sequence and length, but also protein function required for insulin secretion, survival and integrity.

## 4. Material and Methods

### 4.1. Animals and Study Design

Animals were housed in an 22 ± 2 °C environment with a 12:12 h light:dark cycle and unlimited access to food and water. All animal experiments were approved by the ethics committee of the State Office of Environment, Health and Consumer Protection (Federal State of Brandenburg, Brandenburg, Germany). The approval number is: V3-2347-26-2014. The study was performed with male mice using an established feeding protocol as described previously [1]. Briefly, mice were kept on a carbohydrate-free diet until the age of 18 weeks, followed by two days of a carbohydrate-enriched diet. Finally, animals were sacrificed, and islets of Langerhans were isolated and processed for RNA-seq analysis. 

### 4.2. RNA Extraction and Sequencing

Total RNA from isolated islets (5 animals per group) was extracted using the miRNeasy Micro Kit from (QIAGEN, Hilden, Germany). RNAseq was performed by GATC on a Hiseq4000 system with 125 bp paired-end reads. All raw datasets of RNAseq results on which the conclusions of the paper rely will be publicly available repositories upon acceptance.

### 4.3. Splice Sensitive RT-PCR and qRT-PCR

For the validation of alternative splicing of *Abcc8*, another group of animals (4 NZO and 4 B6-ob/ob) were used for total RNA preparation. Subsequently, cDNA synthesis was conducted with M-MLV reverse transcriptase (M3683, Promega, Fitchburg, WI, USA) and *Abcc8*-specific reverse primer (R- 5′GATCTCCATGTCTGCCAGGT3′). The PCR forward primer located in Exon32 (5′ACCTGGCAGACATGGAGATC3′) and the reverse in exon 35 (5′GATCTCCATGTCTGCCAGGT3′ were used in 30 cycles. PCR products were determined with capillary electrophoresis by using the QIAxcel machine (QIAGEN, Hilden, Germany) according to the manufacturer’s instructions. Gene expression of *Srrm4* and *Abcc8* was measured by RT-qPCR and normalized to housekeeping genes *Ppia* and *Eef2* using specific probes (Taqman, ThermoFisher SCIENTIFIC, Waltham, MA, USA and Integrated DNA Technologies IDT, Coralville, IA, USA).

### 4.4. Bioinformatic Analysis

RNAseq data analysis for differential gene expression was performed using salmon and DESeq2. Alternative splicing analysis was performed using Whippet, and downstream processing was conducted using custom Python scripts. The ENSEMBL mm10 annotation was used in all processing steps, the list of microexons from Irimia et al. [27] was ported from mm9 using the liftOver tool. ggSashimi was used for visualization of Sashimi plots. For identification of SNPs between B6-ob/ob and NZO and calculation of SNP frequencies, the annotation from Sanger was used. Rmaps2 was used for the identification of enriched RBP motifs against a background of all alternative mouse exons. Kyoto Encyclopedia of Genes and Genomes (KEGG) analysis was performed by using DAVID. Network analysis was obtained with Ingenuity Pathway Analysis (IPA) (QIAGEN, Hilden, Germany).

## Figures and Tables

**Figure 1 ijms-22-08597-f001:**
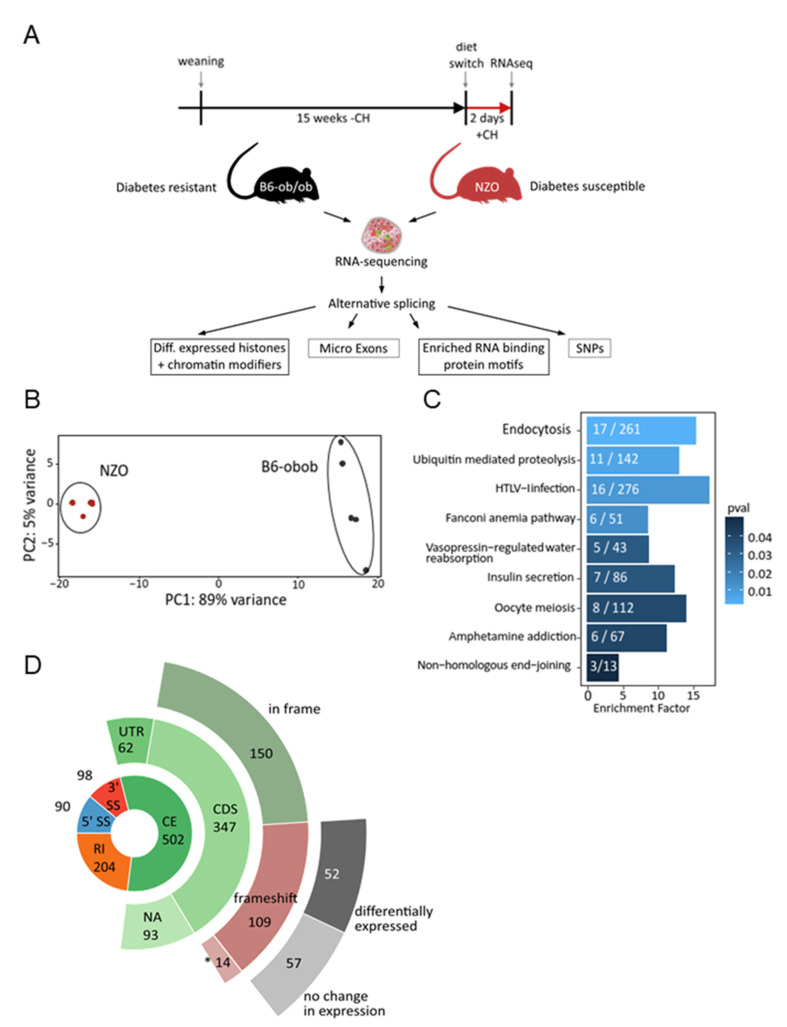
Alternative splicing in islets of NZO and B6-ob/ob mice. (**A**) Feeding regimen of B6-ob/ob and NZO mice, study design, and strategy of screening for alternative splicing. RNAseq analysis was performed after 2 days feeding with carbohydrates (+CH). (**B**) Principal component analysis of RNAseq data. (**C**) KEGG-pathway analysis of alternatively spliced genes. White numbers in bars indicate number of genes. (**D**) Donut plots of identified alternative splicing events and their respective consequences. 5′ ss = alternative 5′ splice site, 3′ ss = alternative 3′ splice site, CE = cassette exon, IR = intron retention. UTR = untranslated region, CDS = coding sequence, NA = not annotated. * Fourteen events have multiple differential exons that might rescue the frame shift.

**Figure 2 ijms-22-08597-f002:**
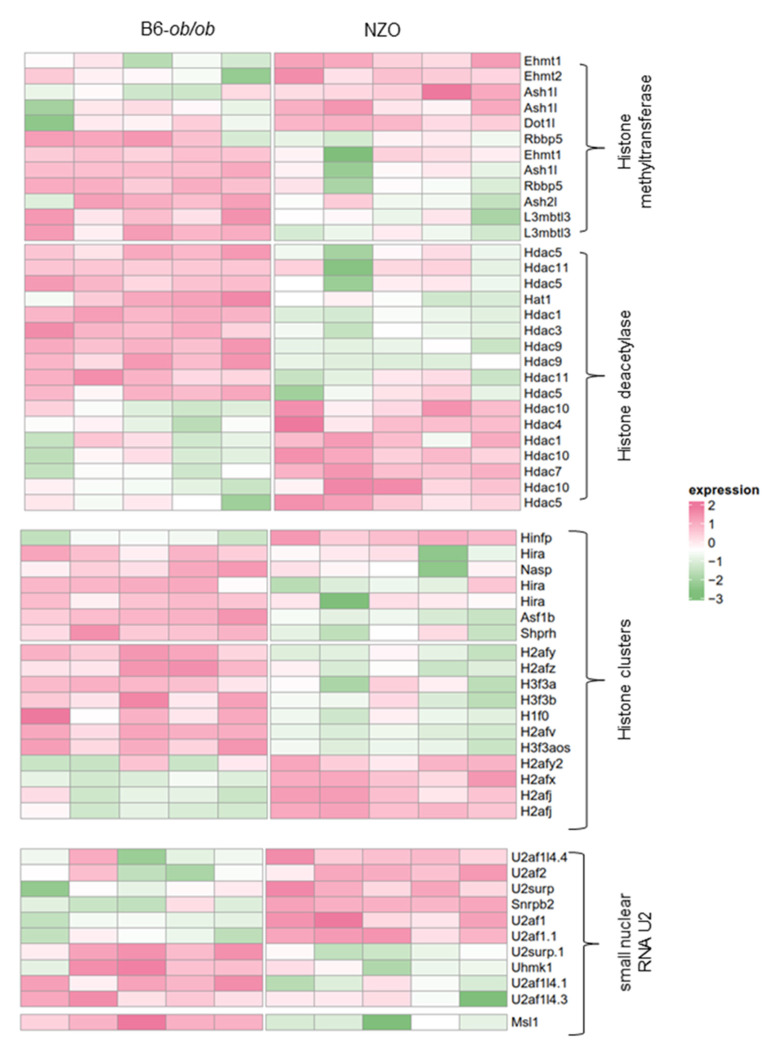
Differential expression of histone and chromatin modifiers. Heatmaps showing expression levels of the indicated genes. Each column represents the individual expression level in islets of B6-ob/ob and NZO mice; each row shows the expression profile of one single transcript with significant differences. Up- and downregulated genes are indicated by light pink and light green signals, respectively; the signal intensity corresponds to the log-transformed scaled magnitude of the individual FPKM (Fragments Per Kilobase Million) levels. Five animals were used per group to perform Welch test.

**Figure 3 ijms-22-08597-f003:**
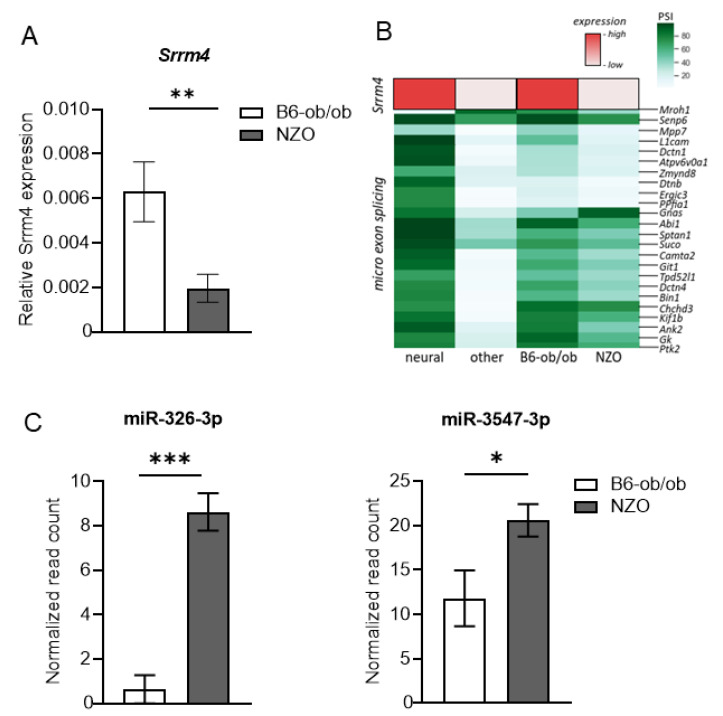
Dysregulated splicing of microexons. (**A**) Relative expression of *Srrm4* in islets of male B6-ob/ob and NZO mice (*n* = 4) analyzed by qRT-PCR. Data are shown as mean ± SD, Student’s *t*-test with Welch’s correction. (**B**) Relative expression of *Srrm4* from our and published data [27]. RNAseq data shown in red. Analysis of genes containing spliced microexons as listed in [27] in different tissues and islets of Langerhans from B6-ob/ob and NZO male mice represented in green. PSI = percent spliced in. (**C**) Elevated expression of miR-326-3p and miR-3547-3p in islets of NZO. Data extracted from miRNA-seq data from [29].* *p* < 0.05; ** *p* < 0.01; *** *p* < 0.001.

**Figure 4 ijms-22-08597-f004:**
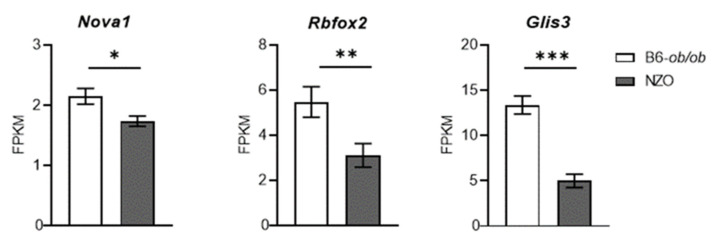
Differential expression of neuron specific splicing factors and the diabetes gene *Glis3*. Expression of *Nova1*, *Rbfox2*, and *Glis3* of islets from B6-ob/ob and NZO mice was detected by RNAseq. * *p* < 0.05; ** *p* < 0.01; *** *p* < 0.001.

**Figure 5 ijms-22-08597-f005:**
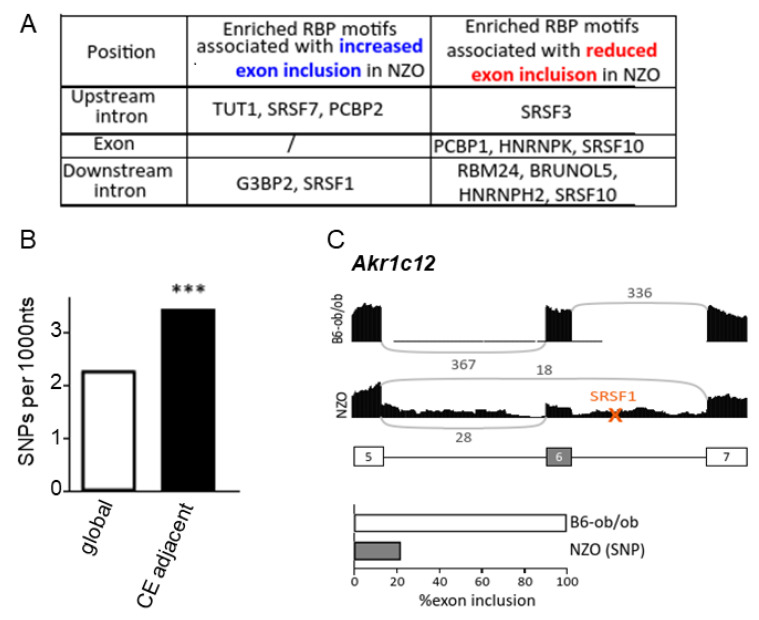
Impact of SNPs on alternative splicing. (**A**) Enrichment of RBP motifs in or close to differentially spliced exons and the predicted consequence for exon inclusion. (**B**) Enrichment of SNPs affecting cassette exon (CE) splicing. Bar graph summarizing the global SNPs frequency vs. SNPs located in close proximity to CE *** *p* ≤ 0.001, chi^2^ test. (**C**) Schematic illustration of a SNP in a SRSF1 binding site downstream of an alternative exon in *Akr1c12*. This polymorphism leads to enhanced skipping in islets from NZO mice. Sashimi plot and quantification are shown below. The numbers above the lines indicate the raw sequencing reads. Orange “X” marks the approximate position of RBP motif-changing SNPs. A simplified exon-intron structure is shown below the Sashimi plot. (**D**) Circos plot summarizing results of the gene ontology analysis of genes carrying SNPs in or in proximity to splice sites. Each line relates a gene to the indicated pathway, with each color representing one biological process. For example, green lines depict the connection to small GTPase mediated signal transduction pathway. (**E**) Schematic presentation of the *Abcc8* exons 33 and 34 of splicing in B6-ob/ob (top) and NZO islets (bottom). Illustration of SNPs (highlighted in grey) in proximity of the splice site in the human or mouse B6-ob/ob and NZO sequence. Capital letters indicate exonic bases, small letters indicate intronic bases. Dashed grey line represents exon/intron boundary. (**F**) Sashimi plot of *Abcc8* from B6-ob/ob and NZO islets showing raw sequencing reads above a simplified exon-intron structure. Orange “X” marks the approximate position of SNPs in the NZO sequence. (**G**) Results of splice sensitive RT-PCR with RNA from islets of 4 B6-ob/ob and NZO mice separated on a QIAxcel machine. Green lines indicate the alignment marker. Arrows above simplified exon structure indicate primer localization.

**Table 1 ijms-22-08597-t001:** In silico analysis of genes with differential splicing events in NZO islet and described in T2D GWAS analysis.

Gene	Type	Psi_NZO	Psi_obob	DeltaPsi	Probability	Category	Log2FC	*p*-Value	Expression Change	#snps Intronic Upstream	#snps 3 Prime ss Adjacent	#snps Exonic	#snps 5 Prime ss Adjacent	#snps Intronic Downstream	#snps Total	#CG	Phenotypes
*Abcc8*	CE	0.039	0.999	–0.959	1	strong exclusion	–0.14	6.35 × 10^–3^	not significant	0	0	0	1	0	1	6	Impaired glucose tolerance
*Tcf7l2*	CE	0.079	0.629	–0.551	0.98	strong exclusion	–0.71	1.52 × 10^–5^	significant	1	0	0	0	1	2	3	Incomplete penetrance (HOM), decrease total body fat
*Pbx4*	CE	0.237	0.647	–0.410	0.95	moderate exclusion	–1.29	7.82 × 10^–4^	significant	1	0	0	0	5	6	3	Not tested
*Nudt6*	CE	0.104	0.488	–0.385	0.952	moderate exclusion	0.13	3.01 × 10^–1^	not significant	11	2	0	1	5	19	0	Not tested
*Tpcn2*	CE	0.562	0.863	–0.301	0.909	moderate exclusion	–1.26	5.51 × 10^–11^	significant	1	0	0	0	7	8	3	Increased circulating triglyceride level
*Zfp945*	CE	0.143	0.424	–0.281	0.919	moderate exclusion	–0.66	1.45 × 10^–5^	significant	0	0	0	0	2	2	3	Not tested
*Myt1l*	CE	0.746	0.984	–0.238	0.995	moderate exclusion	–0.05	5.68 × 10^–1^	not significant	6	3	3	1	14	27	0	Decreased circulating glycerol level
*Aktip*	CE	0.217	0.061	0.155	0.995	moderate inclusion	–0.02	7.07 × 10^–1^	not significant	1	0	1	0	4	6	0	Decreased subcutaneous adipose tissue amount, decreased total body fat amount, decreased body weight
*Fbrsl1*	RI	0.295	0.114	0.180	0.982	moderate inclusion	0.13	1.69 × 10^–1^	not significant	1	0	1	0	2	4	15	Preweaning lethality, incomplete penetrance (HOM)
*Nrxn1*	CE	0.908	0.568	0.340	0.975	moderate inclusion	–2.29	3.09 × 10^–61^	significant	1	0	0	0	0	1	1	Decreased fasting circulating glucose level, impaired glucose tolerance, decreased circulating HDL cholesterol level, decreased body fat amount, decreased circulating cholesterol level
*Nrxn1*	CE	0.917	0.569	0.348	0.979	moderate inclusion	–2.29	3.09 × 10^–61^	significant	1	0	0	0	1	2	0	Decreased fasting circulating glucose level, impaired glucose tolerance, decreased circulating HDL cholesterol level, decreased body fat amount, decreased circulating cholesterol level
*Arl15*	CE	0.613	0.103	0.510	0.995	strong inclusion	–0.17	3.31 × 10^–2^	not significant	5	1	0	0	4	10	1	Preweaning lethality, complete penetrance (HOM)

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
