# Peer review of "Enriched Alternative Splicing in Islets of Diabetes-Susceptible Mice"

_ijms, 2021, doi:10.3390/ijms22168597_

Round 1

Reviewer 1 Report

This is an interesting study. This study demonstrated that the repression of Srrm4 expression is mediated via a higher expression of miR-326-3p and miR-3547-3p in islets of diabetic mice. This study suggests that an altered splicing pattern in islets of diabetes-susceptible mice may contribute to an elevated T2D risk.

This study has a very small sample size. This study only performed bioinformatic analysis in 5 mice per group.  

Author should also add in vitro experiments to confirm bioinformatic findings.

Author Response

Reviewer 1

We would like to thank reviewer 1 very much for evaluating the revision (ijms-1280680) for his constructive comments and suggestions.

Point 1. This study has a very small sample size. This study only performed bioinformatic analysis in 5 mice per group.  

Response 1.As the study is based on the comparison of two mouse strains of which each is genetically identical, a sample size of 5 per group is appropriate. Similar RNAseq based studies on islets e.g. of db/+ in comparison to db/db mice were performed even with only 3 samples per genotype [1]. Also the group of Dr. Drucker used 5-6 samples for RNAseq of islets [2]. These are just two of several examples. We therefore think that the sample size is big enough, which is further underlined by the principle component analysis in Fig. 1B that shows the variation between the two genotypes but low variation within one group.

Point 2. Author should also add in vitro experiments to confirm bioinformatic findings.

Response 2. The reviewer is right, most of the analyses are bioinformatic studies, which provide broad information on alternative splicing events that occur in a different manner in islets of diabetes-prone and diabetes-resistant mice. It is impossible to validate all these events by in vitro analysis. However, we experimentally validated exon skipping of Abcc8 and show these results in Fig. 3.

References

  1. Neelankal John, A.; Ram, R.; Jiang, F. X., RNA-Seq Analysis of Islets to Characterise the Dedifferentiation in Type 2 Diabetes Model Mice db/db. Endocr Pathol 2018, 29, (3), 207-221.
  2. Kaur, K. D.; Wong, C. K.; Baggio, L. L.; Beaudry, J. L.; Fuchs, S.; Panaro, B. L.; Matthews, D.; Cao, X.; Drucker, D. J., TCF7 is not essential for glucose homeostasis in mice. Mol Metab 2021, 48, 101213.

Reviewer 2 Report

This manuscript presents an analysis of alternative splicing in pancreatic islets of mice that are susceptible to diabetes compared to controls. These mice with different genetic backgrounds were fed with a carbohydrate diet as diabetogenic stimulus. By using RNA-seq and few downstream analyses, authors identified many differential alternative splicing events that could be associated to diabetes, including microexons, as well as the expression changes of chromatin modifiers and splicing factors. Last, authors analysed the alternative splicing events associated to SNPs that are divergent from either mouse strain, and some of these may explain why these mice respond differently to the carbohydrate challenge. Overall the study is ok, yet there are few minor corrections to be introduced before recommending this for publication in IJMS:

  1. First, the connection between SNPs and particular alternative splicing events of the gene that contains them is largely overstated. The connection between a SNP variant and a splicing phenotype cannot be attributed just by looking at two genotypes, because these mice strains diverge in many other SNPs that could cause the splicing change. For instance, another SNP in a splicing factor might cause or contribute to the alternative splicing changes. Instead, to connect a SNP with a splicing outcome needs analysis of many genotypes and statistical tests, which are beyond the scope of this study. Nevertheless, this limitation should be discussed and the conclusions of this part largely toned down.
  2. The result of the Abcc8 SNP indeed is far from clear. This SNP changes a G for an A at position +3 of a 5’ splice site. Few papers that are dated back more than a decade show that 5’ss +3A is stronger than +3G in a context dependent manner. However, the +3A in NZO mice results in skipping of the exon, while this exon is included in other mice with +3G. This counterintuitive result suggests that the SNP is not responsible of the splicing change in Abcc8, or alternatively that for this particular 5’ splice site, a +3G is stronger than a +3A. This should be discussed and citations added.
  3. SRRM4 expression in the islets of these mice is correlated to the splicing of microexons. For this it would be good to show changes in expression of SRRM4 protein by western blotting in addition to changes in RNA by RNA-seq (Figure 3).
  4. In Figure 1D donut plot, it would be meaningful to indicate the number of alternative splicing events in addition to the percentages.
  5. Last, the text should be improved for a better readability throughout the whole manuscript.

Author Response

Reviewer 2

First of all, we would like to thank the reviewer for her/his positive comments.

  1. First, the connection between SNPs and particular alternative splicing events of the gene that contains them is largely overstated. The connection between a SNP variant and a splicing phenotype cannot be attributed just by looking at two genotypes, because these mice strains diverge in many other SNPs that could cause the splicing change. For instance, another SNP in a splicing factor might cause or contribute to the alternative splicing changes. Instead, to connect a SNP with a splicing outcome needs analysis of many genotypes and statistical tests, which are beyond the scope of this study. Nevertheless, this limitation should be discussed and the conclusions of this part largely toned down.

Response.1. We thank Reviewer #2 for this comment and agree that linking SNPs directly to alternative splicing events is overstated. We are aware that the study presented is not a global analysis. However, we are convinced that comparing the afore mentioned genotypes has the power to identify disease causing or promoting factors as it was shown earlier by identifying diabetes genes that were even translated to diabetic human patients (Kluth et al 2014; Kluth et al 2019). Nevertheless, we have weakened the description and interpretation of this data significantly and mentioned the limitation of the sole SNP analysis in the discussion where we state, Whether the SNPs identified and discussed here directly affect the detected differences in splicing, needs further proof. It is also possible that SNPs located not in splicing sites but in genes encoding for splicing factors impact splicing.’ Lines 276-279 and, This observation is limited to bioinformatic analysis of RNAseq data from only two different mouse strains and needs to be further proven. Nevertheless, the comparison of two mouse strains with different genotypes was already successful to identify a single SNP that has an impact on metabolic health in another context (Schwerbel et al, 2020)’. Lines 368-371.

  1. The result of the Abcc8 SNP indeed is far from clear. This SNP changes a G for an A at position +3 of a 5’ splice site. Few papers that are dated back more than a decade show that 5’ss +3A is stronger than +3G in a context dependent manner. However, the +3A in NZO mice results in skipping of the exon, while this exon is included in other mice with +3G. This counterintuitive result suggests that the SNP is not responsible of the splicing change in Abcc8, or alternatively that for this particular 5’ splice site, a +3G is stronger than a +3A. This should be discussed and citations added.

Response.2. We agree with Reviewer #2 that the result for Abcc8 splicing is contra intuitive. Since a G to A mutation at the +3 position promotes base pairing with the U1 snRNA (Yeo et al, 2004, PMID: 15285897) it should support exon definition (Talerico et al, 1990, PMID: 2247057). However, disease associated G to A mutations were found in other cases and were suggested to act in a context dependent manner (Roca et al, 2008, PMID: 18032726).While we further cannot rule out that other mutations or differences in the trans acting environment are responsible for the observed splicing change, we consider it more likely that the complex recognition of the two consecutive exons is inhibited by changing the second 5’ splice site. We adapted the respective text clarifying the effect of the mutation and discuss other modes of regulation in lines 310-317.

  1. SRRM4 expression in the islets of these mice is correlated to the splicing of microexons. For this it would be good to show changes in expression of SRRM4 protein by western blotting in addition to changes in RNA by RNA-seq (Figure 3).

Response.3. We agree, it would be even more convincing to validate the differences of SRRM4 expression on protein level. However, an anti-SRRM4 antibody which is commercially available did not work, because it detected a large number of unspecific signals that did not vanish even after enrichment of nuclear fractions. We therefore did not trust the specificity of this antibody. Fig. 3A actually shows the result of qRT-PCR data obtained from isolated islets from 4 mice per genotype. We now highlight this in the text as well as in the figure legend.

  1. In Figure 1D donut plot, it would be meaningful to indicate the number of alternative splicing events in addition to the percentages.

Response.4. We thank Reviewer #2 for this suggestion to improve Fig. 1D. For more clarity we now show absolute numbers instead of percentages and adapted the text in the respective result section (Lines 141-150).

  1. Last, the text should be improved for a better readability throughout the whole manuscript.

Response.5. We went through the text, corrected mistakes and tried to improve readability.